# GRAPH DEFORMER NETWORK

## ABSTRACT

Convolution learning on graphs draws increasing attention recently due to its potential applications to a large amount of irregular data. Most graph convolution methods leverage the plain summation/average aggregation to avoid the discrepancy of responses from isomorphic graphs. However, such an extreme collapsing way would result in a structural loss and signal entanglement of nodes, which further cause the degradation of the learning ability. In this paper, we propose a simple yet effective graph deformer network (GDN) to fulfill anisotropic convolution filtering on graphs, analogous to the standard convolution operation on images. Local neighborhood subgraphs (acting like receptive fields) with different structures are deformed into a unified virtual space, coordinated by several anchor nodes. In space deformation, we transfer components of nodes therein into affinitive anchors by learning their correlations, and build a pseudo multi-granularity plane calibrated with anchors. Anisotropic convolutional kernels can be further performed over the anchor-coordinated space to well encode local variations of receptive fields. By parameterizing anchors and stacking coarsening layers, we build a graph deformer network in an end-to-end fashion. Theoretical analysis indicates its connection to previous work and shows the promising property of isomorphism testing. Extensive experiments on widely-used datasets validate the effectiveness of the proposed GDN in node and graph classifications.

## 1    INTRODUCTION

Graph is a flexible and universal data structure consisting of a set of nodes and edges, where node can represent any kind of objects and edge indicates some relationship between a pair of nodes. Research on graphs is not only important in theory, but also beneficial to in wide backgrounds of applications. Recently, advanced by the powerful representation capability of convolutional neural networks (CNNs) on grid-shaped data, the study of convolution on graphs is drawing increasing attention in the fields of artificial intelligence and data mining. So far, Many graph convolution methods (Wu et al., 2017; Atwood & Towsley, 2016; Hamilton et al., 2017; Velickovic et al., 2017) have been proposed, and raise a promising direction.

The main challenge is the irregularity and complexity of graph topology, causing difficulty in constructing convolutional kernels. Most existing works take the plain summation or average aggregation scheme, and share a kernel for all nodes as shown in Fig. 1(a). However, there exist two non-ignorable weaknesses for them: i) losing the structure information of nodes in the local neighborhood, and ii) causing signal entanglements of nodes due to collapsing to one central node. Thereby, an accompanying problem is that the discriminative ability of node representation would be impaired, and further non-isomorphic graphs/subgraphs may produce the same responses.

Contrastively, in the standard convolutional kernel used for images, it is important to encode the variations of local receptive fields. For example, a $3 \times 3$ kernel on images can well encode local variations of $3 \times 3$ patches. An important reason is that the kernel is anisotropic to spacial positions, where each pixel position is assigned to a different mapping. However, due to the irregularity of graphs, defining and operating such an anisotropic kernel on graphs are intractable. To deal with this problem, Niepert et al. (Niepert et al., 2016) attempted to sort and prune neighboring nodes, and then run different kernels on the ranked size-fixed nodes. However, this deterministic method is sensitive to node ranking and more prone to being affected by graph noises. Furthermore, some graph convolution methods (Velickovic et al., 2017; Wang et al., 2019) introduce an attention mechanism to learn the importances of nodes. Such methods emphasize on mining those significant struc-

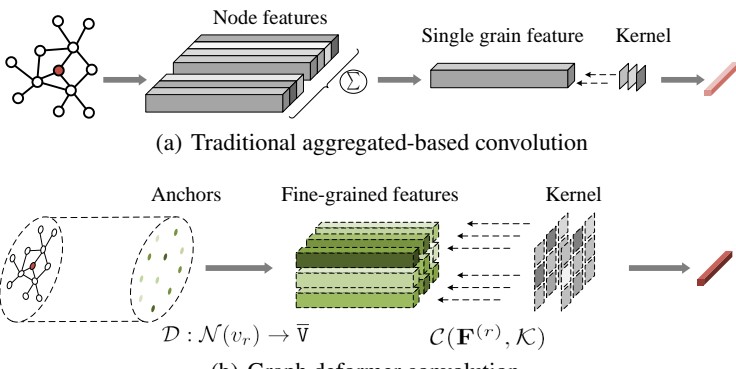

(a) Traditional aggregated-based convolution

(b) Graph deformer convolution

Figure 1: An illustration of ours vs the previous convolution. The red node is a reference node. (a) In traditional graph convolution, the convolution kernel is shared for all nodes due to the plain aggregation over all nodes in the neighborhood. (b) In our method, the irregular neighborhood is deformed into a unified anchor space, which is a pseudo-grid shape, and then the anisotropic convolution kernel is used to encode the space variations of deformable features.

tures/features rather than designing anisotropic convolution kernels, so they cannot well represent local variations of structures in essence.

In this work, we propose a novel yet effective graph deformer network (GDN) to implement anisotropic convolutional filtering on graphs as shown in Fig. 1(b), exactly behaving like the standard convolution on images. Inspired by image-based convolution, we deform local neighborhoods of different sizes into a virtual coordinate space, implicitly spanned by several anchor nodes, where each space granularity corresponds to one anchor node. In order to perform space transformation, we define the correlations between neighbors and anchor nodes, and project neighboring nodes into the regular anchor space. Thereby, irregular neighborhoods are deformed into the anchor-coordinated space. Then, the image-like anisotropic convolution kernels can be imposed on the anchor-coordinated plane, and local variations of neighborhoods can be perceived effectively. Due to the importance of anchors, we also deform anchor nodes with adaptive parameters to match the feature space of nodes. As anisotropic convolution kernels are endowed with the fine-grained encoding ability, our method can better perceive subtle variations of local neighborhood regions as well as reduce signal confusion. We also show its connection to previous work, and theoretically analyze the stronger expressive power and the satisfactory property of the isomorphism test. Extensive experiments on graph/node classification further demonstrate the effectiveness of the proposed GDN.

## 2 OUR APPROACH

In this section, we elaborate on the proposed graph deformer method. Below we first give an abstract formulation for our method and then elaborate on the details.

Denote $\mathcal{G} = (\mathcal{V}, \mathcal{E})$ as an undirected graph , where $\mathcal{V}$ represents a set of nodes with $|\mathcal{V}| = n$ and $\mathcal{E}$ is a set of edges with $|\mathcal{E}| = e$. According to the link relations in $\mathcal{E}$, the corresponding adjacency matrix can be defined as $\mathbf{A} \in \mathbb{R}^{n \times n}$. And $\mathbf{X} \in \mathbb{R}^{n \times d}$ is the feature matrix. To state conveniently, we use $\mathbf{X}_{i \cdot}$ or $\mathbf{x}_i$ to denote the feature of the $i$-th node. Besides, for a node $v_i$, the first-order neighborhood consists of nodes directly connected to $v_i$, which is denoted as $\mathcal{N}_{v_i}^1 = \{v_j | (v_j, v_i) \in \mathcal{E}\}$. Accordingly, we can define $s$-order neighborhood $\mathcal{N}_{v_i}^s$ as the set of $s$-hop reachable nodes.

### 2.1 A BASIC FORMULATION

Given a reference node $v_r$ in graph $\mathcal{G}$, we need to learn its representation based on the node itself as well as its contextual neighborhood $\mathcal{N}_{v_r}$. However, the irregularity causes difficulty in designing anisotropic spatial convolution. To address this problem, we introduce anchor nodes to deform the neighborhood. All neighboring nodes are calibrated into a pseudo space spanned by anchors. We

denote the set of anchor nodes by $\overline{V} = \{\overline{v}_0, \overline{v}_1, ..., \overline{v}_{m-1}\}$. The convolution on $\mathcal{N}_{v_r}$ is formulated as:

$$\widetilde{\mathbf{x}}_r = (\mathcal{G} * f)(v_r) = \mathcal{C}(\mathbf{F}^{(r)}, \mathcal{K}), \tag{1}$$

$$\mathbf{F}_i^{(r)} = \sum_{v_t \in \mathcal{N}_{v_r}} \mathcal{D}_{v_t \to \overline{v}_i}(\overline{\mathbf{x}}_i, \mathbf{x}_t, \Theta), \tag{2}$$

where

- $\overline{v}_., \overline{\mathbf{x}}_.$: an anchor node and a pseudo coordinate vector (a.k.a. feature vector). Please see Section 2.2 for anchor generation.

- $\mathcal{D}$: the deformer function. It transforms node $v_t$ into a virtual coordinate space spanned by anchors. $\Theta$ is the deformer parameter to be learned. Please see Section 2.3.1 for details.

- $\mathbf{F}^{(r)} \in \mathbb{R}^{m \times d}$: the deformed multi-granularity feature from the neighborhood of node $v_r$. Each granularities $\mathbf{F}_i^{(r)}$ corresponds to an anchor node $\overline{v}_i$.

- $\mathcal{C}, \mathcal{K}$: the anisotropic convolution operation on anchor space and convolution kernel. $\mathcal{G} * f$ represents filter $f$ acting on graph $\mathcal{G}$. The relationship between anchor nodes can be built by some metrics such as Cosine distance, and anchor nodes may be format as a pseudo 2-D grid just like the patch in images. Please see the details in Section 2.3.2.

## 2.2 ANCHOR GENERATION

Anchor nodes are crucial to the graph convolution process, because neighborhood regions are unitedly calibrated with them. Rigid anchors will not adapt to the variations of the feature space during convolution learning. Thus we choose to optimize anchor nodes as one part of the entire network learning. In the beginning, we cluster some nodes randomly sampled from the graph as initial anchors. When enough anchors cover the space of neighborhood nodes, the anchors can be endowed with a strong expressive ability to encode neighborhoods like a code dictionary. Formally, we use the K-means clustering to generate initial anchors,

$$\overline{\mathcal{V}} \leftarrow \text{Clustering } \{(v_i, \mathbf{x}_i)|v_i \in \mathcal{V}_{\text{sampling}}\}, \tag{3}$$

where $\mathcal{V}_{\text{sampling}}$ are the sampled node set, in which each node is randomly sampled from the graph, $\overline{\mathcal{V}} = \{(\overline{v}_k, \overline{\mathbf{x}}_k)\}|_{k=0}^{m-1}$ is the initial anchor set generated by clustering, in which $\overline{v}_k$ represents $k$-th anchor node and $\overline{\mathbf{x}}_k$ represents its feature vector, $m$ is the number of anchor nodes. Note that when given anchor nodes, the response of our method will be invariant however to permute nodes of one graph during the training stage as well as testing stage. The clustering algorithm might affect the final anchors due to random sampling for initialization, but it cannot affect the property of permutation invariance, which just like random initialization on the network parameters.

A larger $m$ could increase the expression capacity of anchors, but causes more redundancy and a larger computational cost. Due to the sparsity of graphs, in practice, several anchors are sufficient to encode each neighborhood region. To better collaborate with node/feature variations during graph convolution learning, we transform the initial anchors into a proper space by parameterizing them:

$$\overline{\mathbf{a}}_k = \text{ReLU } (\mathbf{W}_A \overline{\mathbf{x}}_k + \mathbf{b}_A), \quad k = 0, 1, \cdots, m - 1, \tag{4}$$

where $\mathbf{W}_A, \mathbf{b}_A$ are the learnable parameters, and ReLU is the classic activation function. Besides, other flexible multi-layer networks may be selected to learn deformable anchors.

## 2.3 DEFORMER CONVOLUTION

### 2.3.1 SPACE TRANSFORMATION

Now we define the deformer function $\mathcal{D}$ in Eqn. (2), which transforms neighborhood nodes to the anchor space. For each node $v_j \in \mathcal{N}_{v_r}$, we derive the anchor-related feature (also query feature) and value feature vectors as

$$\mathbf{q}_j = \text{ReLU } (\mathbf{W}_Q \mathbf{x}_j + \mathbf{b}_Q), \quad j = 0, 1, \cdots, n_r - 1, \tag{5}$$

$$\mathbf{u}_j = \text{ReLU } (\mathbf{W}_U \mathbf{x}_j + \mathbf{b}_U), \quad j = 0, 1, \cdots, n_r - 1, \tag{6}$$

where $\mathbf{W}_Q, \mathbf{W}_U$ are the learnable weight matrices, and $\mathbf{b}_Q, \mathbf{b}_U$ are the biases. The query feature $\mathbf{q}_j$ indicates how to transform $v_j$ to the anchor space by interacting with anchors, and the value vector $\mathbf{u}_j$ is the transformable component to the anchor space.

For the neighborhood $\mathcal{N}_{v_r}$, the correlation to anchors defines a set of weights $\alpha = \{\alpha_{1,1}, \cdots, \alpha_{1,m}, \cdots, \alpha_{n_r,1}, \cdots, \alpha_{n_r,m}\}$, which measures the scores of all nodes within the neighborhood projected onto the directions of anchor nodes. Formally,

$$\alpha_{j,k} = \frac{\exp(\langle \mathbf{q}_j, \overline{\mathbf{a}}_k \rangle)}{\sum_{k'} \exp(\langle \mathbf{q}_j, \overline{\mathbf{a}}_{k'} \rangle)}, \quad k' = 0, 1, \cdots, m-1, \tag{7}$$

where $\langle \cdot, \cdot \rangle$ denotes the inner production, then normalization is done by softmax function. $\alpha_{j,k}$ may be viewed as the attention score of the node $v_j$ w.r.t. the anchor $\overline{v}_k$. After obtaining the attention score, the irregular neighborhood can be transformed into the anchor-coordinated space,

$$\widetilde{\mathbf{u}}_k = \sum_j \alpha_{j,k} \mathbf{u}_j, \quad j = 0, 1, \cdots, n_r - 1. \tag{8}$$

The deformed components are accumulated on each anchor, and form the final deformed features. Thus, any neighborhood with different sizes can be deformed into the virtual normalized space coordinated by anchors. In experiment, for simplicity, the query feature and value feature are shared with the same parameters in Eqns. (5) and (6).

### 2.3.2 ANISOTROPIC CONVOLUTION IN THE ANCHOR SPACE

Afterward, $s$-hop neighborhood of node $v_r$ is deformed into the size-fixed anchor space, i.e., $\mathcal{N}_{v_r}^s \rightarrow \{\widetilde{\mathbf{u}}_0, \widetilde{\mathbf{u}}_1, \cdots, \widetilde{\mathbf{u}}_{m-1}\}$. The anisotropic graph convolution can be implemented by imposing different mapping on each anchor as

$$\widetilde{\mathbf{x}}_r^{(s)} = \text{ReLU}\left(\sum_i \mathcal{K}_i^{\mathsf{T}} \widetilde{\mathbf{u}}_i + \mathbf{b}\right), \quad i = 0, 1, \cdots, m-1, \tag{9}$$

where $\widetilde{\mathbf{x}}_r^{(s)} \in \mathbb{R}^{d'}$, the matrix $\mathcal{K}_i$ is a $d \times d'$ weight parameter imposed on the features w.r.t. an anchor, and $\mathbf{b}$ is the bias vector.

In the convolution process, different filter weights are imposed on different features of anchor nodes, which is an anisotropic filtering operation. For an intuitive description, we assume the simplest case of 2-D space, which of course can be extended higher dimension. Assuming in 2-D, we project all neighborhood nodes onto anchor nodes, then employ different filters on different anchor nodes in 2-D plane, which likes the standard convolution, so called anisotropic convolution. In contrast to the traditional aggregation method, the deformer convolution has two aspects of advantages: i) well preserving structure information and reducing signal entanglement; ii) transforming different-sized neighborhoods into the size-fixed anchor space to well advocate anisotropic convolution like the standard convolution on images.

### 2.3.3 MULTI-SCALE EXTENSION

Intuitively, the first-order neighborhood is necessary to be used for node aggregation, because it indicates that two nodes linked by an edge are always similar. However, real-world graphs are often so sparse, and there exist many nodes that are similar to each other but not linked by direct edges. The first-order neighborhood alone is not sufficient for extracting useful features and preserving the structural information. It is natural to incorporate higher-order proximity to capture more information. Generally, second-order information is sufficient as most works (Tang et al., 2015; Wang et al., 2016). Higher-order information can also be considered, but the computational complexity will increase. It can be understood as a trade-off of expression ability and computational complexity. In this paper, we consider both first-order and second-order neighborhoods. Specifically, we deform both first-order and second-order neighborhoods into feature space represented by anchor nodes, and convolve over them respectively. Then the learned different neighborhood representations and the original node feature are concatenated as the final filtering response,

$$\widetilde{\mathbf{x}}_r \leftarrow [\mathbf{x}_r; \widetilde{\mathbf{x}}_r^{(1)}; \widetilde{\mathbf{x}}_r^{(2)}], \tag{10}$$

where $\widetilde{\mathbf{x}}_r$ denotes the convolution response on the $s$-order neighborhood $\mathcal{N}_{v_r}^s$ of node $v_r$. Further, we can stack multiply layers to extract more robust features on larger receptive fields.

## 2.4 COARSENING

Graph coarsening can not only reduce the computational cost but also enlarge the receptive field to learn abstract features like image pooling. Below we simply introduce this operation used here.

**Node Classification**. We do not need to remove nodes, and thus name it as pooling. The pooling is node-wise diffusion on a local region, and be performed over multi-scale neighborhoods. The pooling over $S$ scale neighborhoods w.r.t. the reference node $v_r$ is

$$\mathcal{P}(\mathcal{G}(v_r)) = \mathcal{P}(\{\mathbf{x}_j | v_j \in \mathcal{N}_{v_r}^s, s = 1, \cdots, S\}), \tag{11}$$

where the pooling $\mathcal{P}$ is usually defined as "max" or "mean". In practice, their performance has little difference in graph convolution, so we choose the mean operation in our experiments.

**Graph Classification**. We employ the graph cut method used in (Jiang et al., 2019) to partition an entire graph into several subgraphs. During graph coarsening, a binary cluster matrix $\mathbf{Z} \in \mathbb{R}^{n \times c}$ is obtained, where only one element in each row is non-zero, i.e., $Z_{ic} = 1$ when the vertex $v_i$ falls into the cluster $c$. Then the adjacent matrix and feature matrix of the input graph are transformed into

$$\mathbf{A}' \leftarrow \mathbf{Z}^\mathsf{T} \mathbf{A} \mathbf{Z}, \quad \mathbf{X}' \leftarrow \mathbf{Z}^\mathsf{T} \otimes \widetilde{\mathbf{X}}, \tag{12}$$

where $\otimes$ represents a max operation. The output can be used as the input of the next convolutional layer. Then the graph convolution and coarsening can be alternatingly stacked into a deep network.

## 2.5 LOSS FUNCTION

For node and graph classifications, the final convolution output is denoted as $\widehat{\mathbf{Y}}$ after feeding several network layers forward. Then we use the cross-entropy loss on the training set $\mathbf{D}_{tr}$,

$$\mathcal{L} = \frac{1}{|\mathbf{D}_{tr}|} \sum_{v_i \in \mathbf{D}_{tr}} (\mathbf{Y}_{ij} == 1) \ln \widehat{\mathbf{Y}}_{ij}. \tag{13}$$

However, in the scenario of semi-supervised node classification, a main limit is that a small portion of nodes is annotated as the training set. Our aim is to use labeled nodes as well as graph structure to train a model with a good generalization ability. A straightforward way is to add a regularization term to avoid overfitting. To this end, we employ a global consistency constraint through positive pointwise mutual information as used in (Zhuang & Ma, 2018) to regularize the loss function.

## 2.6 COMPUTATIONAL COMPLEXITY

For the computational complexity, we analyze the main module of graph convolution. In one-layer convolution, GCN (Kipf & Welling, 2016) is about $\mathcal{O}(edS + ndd'S)$, where $n, e$ are the numbers of nodes and edges, $S$ is the scale of the neighborhood, and $d, d'$ are the dimensions of the input/hidden layers. For our GDN model, the computational complexity is mainly from two parts, i.e., "Space Transformation" and "Convolution in Anchor Space", which are about $\mathcal{O}(emd^2S)$ and $\mathcal{O}(nmdd'S)$, respectively, where $m$ is the number of anchor nodes. Thus, the total computational complexity is $\mathcal{O}(emd^2S + nmdd'S)$, which is linearly proportional to GCN with the factor $m$ when $d$ and $d'$ have the same order number.

## 3 CONNECTION TO RELATED WORK

In contrast to previous methods (Kipf & Welling, 2016; Xu et al., 2019; Zhuang & Ma, 2018), etc., the way of aggregation is obviously different. GCNs usually bypass the irregularity of graphs by utilizing a weighted-sum aggregation over the neighborhoods, which is an isotropic filter. Our GDN instead firstly transforms the local neighborhoods into a regular anchor space and then performs anisotropic filters on the regular anchor space.

In contrast to Transformer (multi-head attention mechanism) (Vaswani et al., 2017) and its a generalization, graph attention network (Velickovic et al., 2017), we give the following differences. In the Transformer, all center nodes are anchor nodes, and the attention coefficient is computed between each central node and its neighbor nodes. In our GDN, several anchor nodes are initially generated

by K-means Cluster from global nodes which implicitly represents several different directions like a $k \times k$ patch (upper left, upper right, etc.) in images, and the attention coefficient is computed between local neighbors and anchor nodes. Because the anchor nodes are generated from the global graph, all local neighborhoods are projected into a common anchor space, some common property for all local neighborhood can be captured by imposing an anisotropic filter on anchors. In contrast, though multi-head mechanism is used, the Transformer is still locally aggregated. Another weakness of the Transformer is computationally intensive, especially for large graphs.

Our proposed GDN is different form other peer works. P-GNN (You et al., 2019) selects several anchor nodes as position characterization, and concatenates the feature of node with features of anchor nodes, then mean aggregation is employed on different anchor sets followed by a fully-connected transform, which is an isotropic filter. LGCN (Gao et al., 2018) performs feature selection by sorting the top $k$-largest values on each feature dimension, and produces $k$ new nodes (fixed size) for the next filtering process. CapsGNN (Xinyi & Chen, 2018) presents a capsule graph network to learn graph capsules of different aspects from node capsules by utilizing an attention module and routing mechanism. In contrast, our proposed method transforms local neighborhood into an anchor space spanned by several anchor nodes, the filtering is operated in the anchor space. Transforming irregular structures into a regular space also be studied in 3D domain. PointCNN (Li et al., 2018b) learns an X-transformation from the input points and then applies convolution operator on the X-transformed features. KPConv (Thomas et al., 2019) takes radius neighborhoods as input and processes them with weights spatially located by a small set of kernel points. In essence, PointCNN (Li et al., 2018b) leverages the self-attention mechanism to produce fixed-size output for different-size local neighbor regions. KPConv (Thomas et al., 2019) projects neighbor 3D points into 3D kernel points, and this projection is only operated in 3D point space. In contrast, our proposed method focus on the more general graph domain. More related work can be found in Appendix.

## 4 THEORETICAL ANALYSIS

Here, we present a theoretical analysis about the expressive power of several aggregation methods including the mean, sum, and proposed graph deformer operation. Inspired by (Xu et al., 2019), we evaluate them by verifying whether graphs are isomorphic. Then, we give the following propositions.

**Proposition 1.** *There exists a set of network parameters able to graph deformer process can distinguish two non-isomorphic graphs $\mathcal{G}_1$ and $\mathcal{G}_2$, which cannot be distinguished by mean/sum aggregation.*

**Proposition 2.** *The proposed anisotropic graph deformer convolution can be as powerful as the Weisfeiler-Lehman (WL) graph isomorphism test.*

The Proofs of the above two propositions can be found in the Appendix. We can draw the conclusion that the expressive power of the proposed graph deformer network is provably stronger than mean/sum aggregation, which can accomplish an injective as powerful as the Weisfeiler-Lehman (WL) graph isomorphism test.

Table 1: Comparison with state-of-the-art methods on node classification. The number in parentheses $(*)$ denotes the number of convolutional layers in a certain network.

|  | Method | Cora | Citeseer | Pubmed |
|---|---|---|---|---|
| SVM | Planetoid (Yang et al., 2016) | 75.7 | 64.7 | 77.2 |
| Spectral | ChebyNet (Defferrard et al., 2016) | 81.2 | 69.8 | 74.4 |
|  | GCN (Kipf & Welling, 2016) | 81.5 | 70.3 | 79.0 |
|  | DGCN (Zhuang & Ma, 2018) | 83.5 | 72.6 | 80.0 |
| Spacial | MoNet (Monti et al., 2017) | 81.7 | - | 78.8 |
|  | GAT (Velickovic et al., 2017) | 83.0 | 72.5 | 79.0 |
|  | JK-Net (Xu et al., 2018) | $79.71 \pm 0.62$ | $69.03 \pm 0.55$ | $78.17 \pm 0.27$ |
|  | GIN (Xu et al., 2019) | $79.49 \pm 0.65$ | $67.78 \pm 0.89$ | $78.37 \pm 0.29$ |
|  | GDN(1L) | $\mathbf{85.16 \pm 0.47}$ | $73.13 \pm 0.826$ | $79.8 \pm 0.297$ |
|  | GDN(2L) | $84.76 \pm 0.587$ | $\mathbf{73.77 \pm 0.447}$ | $\mathbf{80.77 \pm 0.24}$ |

Table 2: Comparison with state-of-the-art methods on graph classification.

|  | Method | MUTAG | PTC | NCI1 | ENZYMES | PROTEINS | IMDB-BINARY | IMDB-MULTI |
|---|---|---|---|---|---|---|---|---|
| Kernel | WL | $80.72 \pm 3.00$ | $56.97 \pm 2.01$ | $80.13 \pm 0.50$ | $53.15 \pm 1.14$ | $72.92 \pm 0.56$ | $72.86 \pm 0.76$ | $50.55 \pm 0.55$ |
|  | GK | $81.66 \pm 2.11$ | $57.26 \pm 1.41$ | $62.28 \pm 0.29$ | $26.61 \pm 0.99$ | $71.67 \pm 0.55$ | $65.87 \pm 0.98$ | $43.89 \pm 0.38$ |
|  | DGK | $82.66 \pm 1.45$ | $57.32 \pm 1.13$ | $62.48 \pm 0.25$ | $27.08 \pm 0.79$ | $71.68 \pm 0.50$ | $66.96 \pm 0.56$ | $44.55 \pm 0.52$ |
| Feature | FB | $84.66 \pm 2.01$ | $55.58 \pm 2.30$ | $62.90 \pm 0.96$ | $29.00 \pm 1.16$ | $69.97 \pm 1.34$ | $72.02 \pm 4.71$ | $47.34 \pm 3.56$ |
|  | DyF | $88.00 \pm 2.37$ | $57.15 \pm 1.47$ | $68.27 \pm 0.34$ | $33.21 \pm 1.20$ | $75.04 \pm 0.65$ | $72.87 \pm 4.05$ | $48.12 \pm 3.56$ |
| GNN | PSCN | $92.63 \pm 4.21$ | $60.00 \pm 4.82$ | $78.59 \pm 1.89$ | - | $75.89 \pm 2.76$ | $71.00 \pm 2.29$ | $45.23 \pm 2.84$ |
|  | NgramCNN | $94.99 \pm 5.63$ | $68.57 \pm 1.72$ | - | - | $75.96 \pm 2.98$ | $71.66 \pm 2.71$ | $50.66 \pm 4.10$ |
|  | SAEN | $84.99 \pm 1.82$ | $57.04 \pm 1.30$ | $77.80 \pm 0.42$ | - | $75.31 \pm 0.70$ | $71.26 \pm 0.74$ | $49.11 \pm 0.64$ |
|  | IGN | $83.89 \pm 12.95$ | $58.53 \pm 6.86$ | $74.33 \pm 2.71$ | - | $76.58 \pm 5.49$ | $72.0 \pm 5.54$ | $48.73 \pm 3.41$ |
|  | PPGN | $90.55 \pm 8.7$ | $66.17 \pm 6.54$ | $83.19 \pm 1.11$ | - | $77.2 \pm 4.73$ | $72.6 \pm 4.9$ | $50 \pm 3.15$ |
|  | GNTK | $90.0 \pm 8.5$ | $67.9 \pm 6.9$ | $84.2 \pm 1.5$ | - | $75.6 \pm 4.2$ | $76.9 \pm 3.6$ | $52.8 \pm 4.6$ |
|  | CapsGNN | $86.67 \pm 6.88$ | - | $78.35 \pm 1.55$ | $54.67 \pm 5.67$ | $76.28 \pm 3.36$ | $73.10 \pm 4.83$ | $50.27 \pm 2.65$ |
|  | GIC | $94.44 \pm 4.43$ | $\mathbf{77.64 \pm 6.98}$ | $84.08 \pm 1.77$ | $62.50 \pm 5.12$ | $77.65 \pm 3.21$ | $76.70 \pm 3.25$ | $51.66 \pm 3.40$ |
|  | GIN | $89.4 \pm 5.6$ | $64.6 \pm 7.0$ | $82.7 \pm 1.7$ | - | $76.2 \pm 2.8$ | $75.1 \pm 5.1$ | $52.3 \pm 2.8$ |
|  | GDN (3L) | $\mathbf{97.39 \pm 2.65}$ | $75.57 \pm 7.56$ | $\mathbf{86.03 \pm 1.23}$ | $\mathbf{67.5 \pm 6.96}$ | $\mathbf{81.32 \pm 3.09}$ | $\mathbf{79.3 \pm 3.26}$ | $\mathbf{55.2 \pm 4.34}$ |

# 5 EXPERIMENTS

In the section, we carry out extensive experiments to assess the proposed GDN model on both node and graph classification tasks. For node classification, three citation graphs are used: Cora, Citeseer, and Pubmed. For graph classification, we adopt seven datasets to assess our GDN method: MUTAG, PTC, NCI1, PROTEINS, ENZYMES, IMDB-BINARY, and IMDB-MULTI. The details of these datasets and experimental setups can be found in the Appendix.

## 5.1 COMPARISON WITH STATE-OF-THE-ARTS

**Node classification.** We compare the performance of GDN against several baseline works: ChebyNet (Defferrard et al., 2016), GCN (Kipf & Welling, 2016), MoNet (Monti et al., 2017), GAT (Velickovic et al., 2017), DGCN (Zhuang & Ma, 2018), JK-Net (Xu et al., 2018) and GIN (Xu et al., 2019). The accuracies are reported in Table 1, which clearly indicates that our GDN obtains a remarkable improvement. DeepWalk and Planetoid aim to generate effective node embeddings, and our proposed GDN significantly outperforms these two methods. GCN is a first-order approximation of ChebyNet and has realized relatively higher results. Simultaneously, it can be regarded as a special case of Monet, and their classification accuracies are similar. Compared to GCN, our GDN achieves a relatively large gain. We attribute this improvement to the graph deformer convolution. We further compare GDN to GAT, still achieving superior performance on these three datasets. Though GDN utilizes global consistency constraint, there is still a marked improvement compared to DGCN. Compared to recent methods JK-Net and GIN, GDN also obtains a large margin over them. These demonstrate that the proposed GDN method performs well on various graph datasets by building the graph deformer process, where structure variations can be well captured and fine-grained node features can be extracted to enhance the discriminability between nodes.

**Graph classification.** Table 2 shows the results on graph classification. Overall, except for PTC, our GDN approach achieves state-of-the-art performance on all datasets and obtains remarkable improvement. For graph kernel-based methods (WL (Shervashidze et al., 2011), GK (Shervashidze et al., 2009) and DGK (Yanardag & Vishwanathan, 2015)), we can observe the WL kernel can obtain better results on most datasets than GK and DGK. In contrast to WL, the proposed GDN is able to improve by a large margin of 5.9% on NCI1, 14.35% on ENZYMES, 6.44% on IMDB-BINARY, etc. For the feature-based methods (FB (Barnett et al., 2016), DyF (Gomez et al., 2017)), GDN obviously outperforms them. Also, GDN is better than SAEN (Orsini et al., 2017). Recently the GNN-based work (PSCN (Niepert et al., 2016), NgramCNN (Luo et al., 2017), IGN (Maron et al., 2018), PPGN (Maron et al., 2019), GNTK (Du et al., 2019), CapsGNN (Xinyi & Chen, 2018), GIC (Jiang et al., 2019), GIN (Xu et al., 2019)) is superior to traditional machine learning methods. Compared to GIC, the GDN model still achieves superior performances, about 3 percentages on average, although a relatively lower result is gotten on the PTC dataset. This may be attributed to differences in the dataset or less appropriate model parameter settings. Compared with these baseline methods, our GDN can render impressive performance. In summary, the remarkable gains indicate that the proposed GDN is effective to deal with graph classification.

Table 3: Comparison on the scales of neighborhood regions.

| Method | Cora | Citeseer | Pubmed |
|---|---|---|---|
| GDN-$\mathcal{N}^{(0)}$ | $54.95 \pm 2.004$ | $55.7 \pm 1.259$ | $72.94 \pm 1.582$ |
| GDN-$\mathcal{N}^{(0,1)}$ | $82.62 \pm 0.668$ | $72.56 \pm 0.945$ | $80.24 \pm 0.224$ |
| GDN-$\mathcal{N}^{(0,1,2)}$ | $\mathbf{84.76 \pm 0.587}$ | $\mathbf{73.77 \pm 0.447}$ | $\mathbf{80.77 \pm 0.241}$ |

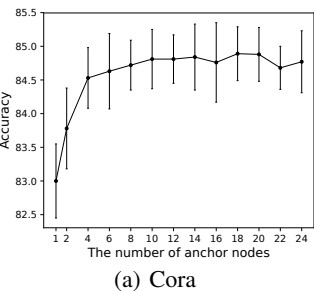

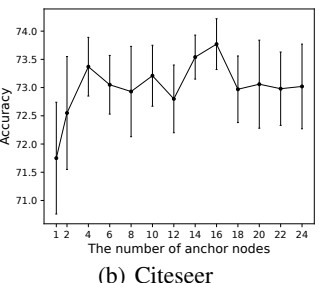

(a) Cora            (b) Citeseer

Figure 2: Comparison on the number $m$ of anchor nodes.

Table 4: The verification of convolutional layer $\mathcal{C}$ and pooling layer $\mathcal{P}$ in our GDN method.

| Method | Cora | Citeseer | Pubmed |
|---|---|---|---|
| GDN w/o $\mathcal{P}$ | $83.97 \pm 0.393$ | $72.53 \pm 0.593$ | $78.86 \pm 0.902$ |
| GDN w/o $\mathcal{C}$ | $82.82 \pm 0.724$ | $71.57 \pm 0.831$ | $78.75 \pm 0.626$ |
| GDN | $\mathbf{84.76 \pm 0.587}$ | $\mathbf{73.77 \pm 0.447}$ | $\mathbf{80.77 \pm 0.241}$ |

## 5.2 ABLATION STUDY

**The scale $s$ of neighborhood region.** The influences of neighborhood scales are reported in Table 3. GDN-$\mathcal{N}^{(0)}$ denotes that only the feature of the node itself is used, GDN-$\mathcal{N}^{(0,1)}$ includes the features of the node itself and first-order neighborhood, and so on. Due to the lack of structural information, we find that the performance of GDN-$\mathcal{N}^{(0)}$ is obviously lower. As more information is considered, the accuracy of GDN-$\mathcal{N}^{(0,1,2)}$ is generally superior to GDN-$\mathcal{N}^{(0,1)}$. This validates the importance of local neighborhood information, which is also a crucial property of traditional CNNs.

**The number $m$ of anchor nodes.** We select the value $m$ in the range $[1, 25]$ to observe the changes of performance. As shown in Fig. 2, when $m = 1$, the accuracies are significantly lower, because only one anchor node is used, which is similar to sum aggregation. When $m = 2$, the performance is also relatively lower, which means 2 anchor nodes are insufficient to capture more variations. Then, the performance is relatively stable with $m$ increasing. The reason should be that, as real-world graph data is rather sparse, e.g., on average about 2 edges for each node in Cora and Citeseer datasets, so a few anchors matching the neighborhood size should be saturated to represent the variations without the information loss.

**Graph convolution and graph coarsening.** We further explore the effectiveness of graph convolution by removing the pooling layer from the GDN model, named as "GDN w/o $\mathcal{P}$". Similarly, "GDN w/o $\mathcal{C}$" means that graph convolution is removed. Table 4 shows the performance on citation datasets. Compared to GDN, both "GDN w/o $\mathcal{P}$" and "GDN w/o $\mathcal{C}$" obtain lower performances. But "GDN w/o $\mathcal{P}$" is better than "GDN w/o $\mathcal{C}$", while they are comparable on the Pubmed dataset. It indicates that the deformer convolution indeed improves the discriminability of nodes. Note that "GDN w/o $\mathcal{C}$" is actually similar to the plain aggregation with the average operation.

**Attention scores $\alpha$.** We visualize correlation scores $\alpha$ between neighborhood nodes and anchors. We respectively select some nodes from Cora and Citeseer datasets as center nodes and compute the scores of their first-order neighbors to anchor nodes. As shown in Fig. 3, for node $V_{20}$ in Cora dataset, the attention score of the neighbor "A" on anchor node $\bar{\mathbf{a}}_4$ is largest while the other four neighbors is close to anchor node $\bar{\mathbf{a}}_1$. For node $v_{153}$ in Citeseer, the neighbors "A" and "C" are more inclined to anchor node $\bar{\mathbf{a}}_6$, while neighbor "D" prefers $\bar{\mathbf{a}}_{12}$, and neighbor "B" is similar in most directions. These neighbors place different emphases on anchor nodes, then different proportions of features are assigned to the directions of these anchors, and an anisotropic convolution is used to extract more fined-grained representation, which is superior to the sum/mean aggregation.

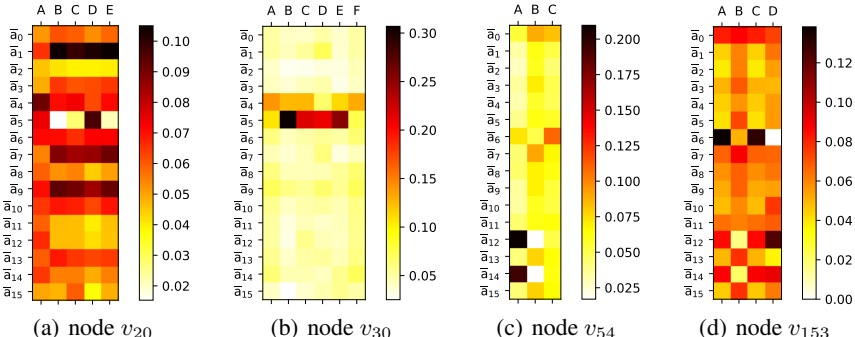

(a) node $v_{20}$     (b) node $v_{30}$     (c) node $v_{54}$     (d) node $v_{153}$

Figure 3: Visualization of attention scores in the first-order neighborhood of nodes $v_{20}$ and $v_{30}$ in the Cora dataset, and $v_{54}$ and $v_{153}$ in the Citeseer dataset. (a) Node $v_{20}$ has 5 neighbors while (b) node $v_{30}$ has 6 neighbors. (c) Node $v_{54}$ has 3 neighbors while (d) node $v_{153}$ has 4 neighbors.

## 6 CONCLUSION

In this work, analogous to the standard convolution on images, we proposed a novel yet effective graph deformer network (GDN) to fulfill anisotropic convolution filtering on graphs. We transformed local neighborhoods with different structures into a unified virtual space, coordinated by several anchor nodes. Anisotropic convolution kernels can thus be performed over the anchor-coordinated space to well encode subtle variations of local neighborhoods. Further, we built a graph deformer network in an end-to-end learning fashion by stacking the deformable convolutional layers as well as the coarsening layers. Our proposed GDN accomplishes significantly better performances on both node and graph classifications. In the future, we will extend the graph deformer method to allow more applications in the real world, such as link prediction, heterogeneous graph analysis, etc.

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
