# OpenReview forum: "Graph Deformer Network"
_ICLR.cc/2021/Conference — Reject_

### Official Review · AnonReviewer2 · 2020-10-28
**Concerns on missing discussion on related 3D literatures and missing ablation study on the number of parameters**

**Rating:** 5
**Confidence:** 4

**Review:**

In order to perform anisotropic convolution on graphs, this paper proposes to project a local neighborhood into a unified virtual space by introducing anchor nodes. A theoretical analysis is provided to show the expressive power of the proposed graph deformer operation on graph isomorphism test. Extensive experiments are performed on several node classification and graph classification datasets.

Pros:

+ The proposed anchor-based anisotropic convolution provides an interesting view of architectural design on graphs;
+  A good theoretical analysis in terms of the expressive power justifies the motivation of the proposed method.

Cons:

- Transforming irregular structures into a regular domain is not new. There is a lack of comparison or discussion with methods in 3D domain such as Point-CNN (Li et al., 2018) and KPConv (Thomas et al., 2019). It seems the proposed method is very similar to KPConv but on graphs instead of point sets;
-The discussion on the number of parameters is missing. In Figure 2, it shows that the best performance is achieved with more than 15 anchor nodes. This means graph deformer network (GDN) will use approximately 15x parameters compared to a regular GNN. It is unclear whether the performance gains come from the usage of more parameters or the proposed anisotropic convolution.

Minor Comment:

* The chosen datasets are relatively small. It would be great if more experiments can be conducted on the Open Graph Benchmark (Hu et al., 2020).


References:
* Yangyan Li, Rui Bu, Mingchao Sun, Baoquan Chen. “PointCNN: Convolution On X -Transformed Points”. arXiv preprint arXiv:1801.07791, 2018.
* Thomas, Hugues, Charles R. Qi, Jean-Emmanuel Deschaud, Beatriz Marcotegui, François Goulette, and Leonidas J. Guibas. "Kpconv: Flexible and deformable convolution for point clouds." In Proceedings of the IEEE International Conference on Computer Vision, pp. 6411-6420. 2019.
* Hu, Weihua, Matthias Fey, Marinka Zitnik, Yuxiao Dong, Hongyu Ren, Bowen Liu, Michele Catasta, and Jure Leskovec. "Open graph benchmark: Datasets for machine learning on graphs." arXiv preprint arXiv:2005.00687 (2020).

Post rebuttal Comments:
Thank the authors for the response. I keep my score as 5.

---

### Official Review · AnonReviewer3 · 2020-10-29
**Recommendation to Reject**

**Rating:** 4
**Confidence:** 4

**Review:**

##########################################################################

Summary:

This work proposes the Graph Deformer Network (GDN), whose key component is the proposed Graph Deformer Convolution (GDC). The GDC is based on the attention mechanism, where a fixed number of query vector comes from a clustering process of some randomly sampled nodes (In my opinion, the q vector in the paper should be named the key vector instead query vector, according to [1]). The attention thus yields a fixed number of ordered vectors as outputs, where an anisotropic convolution can be applied on. Experimental results on both node and graph classification tasks demonstrate the effectiveness of the proposed GDC.

[1] Vaswani et al. "Attention is all you need", NeurIPS 2017.

##########################################################################

Reasons for score:

Overall, I vote for rejection. My major concerns lie in three aspects, as detailed in Cons below: 1) The methods were not clearly described. Many details are lost; 2) The experimental results on graph classification tasks are not convincing; 3) Important studies with the same motivation or similar methods are not discussed.

##########################################################################

Pros:

1. The experimental studies on node classification tasks seem comprehensive and convincing.

2. The motivation is well-explained by comparing graph convolutions with regular convolutions on images.

##########################################################################

Cons:

1. The methods were not clearly described. Many details are lost.
For example, in section 2.1, the f and K in Equation (1) are not explained.
In section 2.2, V_sampling in Equation (3) is not clearly explained. It is not clear what "all neighborhoods" means. Are they just "all nodes"?
In addition, the notation is confusing. How to cluster a tuple? I assume that clustering is performed on feature vectors.
Another key question is that, clustering outputs do not naturally have orders. How do you order those m clusters? This is crucial for graph classification tasks, as the weight parameters K_i is associated with the indices i=0,1,...,m-1. If you don't have certain ways to order the m clusters, the learned parameters may not generalize well.

2. The experimental results on graph classification tasks are not convincing. It is because the proposed GDN uses a different readout function as strong competitors like GIN. The GDN uses Equation (12) plus a convolutional neural network, similar to [2], while GIN simply use sum/avg followed by an MLP. Ablation studies with the same readout function are required to show that the improvement of GDN indeed comes from GDC.

[2] Zhang et al. "An End-to-End Deep Learning Architecture for Graph Classification", AAAI 2018

3. Important studies with the same motivation or similar methods are not discussed, like [3,4].

[3] Gao et al. "Large-Scale Learnable Graph Convolutional Networks", KDD 2018
[4] Zhang et al. "Capsule Graph Neural Network", ICLR 2019

##########################################################################

Questions during rebuttal period:

Please address and clarify the cons above.

##########################################################################

Comments after rebuttal period:

I appreciate the authors' efforts on increasing the clarity of this paper. However, the answers to "How do you order those m clusters?" are not convincing (as well as the responses to R4's 2nd concern, which mentions the same problem). It is a key point in generalization across different graphs. According to the experimental results, I believe there is an implicit mechanism in the proposed method that achieves the ordering, but it is not clearly explained and analyzed. I will keep my score.

---

### Official Review · AnonReviewer1 · 2020-10-31
**The paper proposes a spatial design of an anistropic convolution operator on graphs**

**Rating:** 7
**Confidence:** 5

**Review:**

The authors propose to extend isotropic convolutional operator on graphs to anisotropic one, well suited to encode complex geometric topology of the graphs while providing representations at different levels of granularity. It is achieved by considering a multi scale neighboring system.

One of the challenges in the design of powerful graph convolutional networks is the ability of the network to capture the complex geometric structure of irregular graphs characterized by a strong topological variability where nodes have different number of connections ranging from sparse connectivity to dense one. Moreover, graphs are of variable size.
In contrary to isotropic convolution  which lacks of expressivity and are not capable at leveraging discriminating representations at different direction of information, this paper cope with this problem.

The focus of paper is mainly oriented to the design of anisotropic convolutional operator corroborated by a convincing empirical validation, including different datasets for both node and graph classification. Moreover, the authors provide a theoretical analysis showing that the proposed  Graph Deformer Network under certain conditions, it could distinguish two non-isomorphic graphs (developed in the supplementary part) where most of aggregation operators fail, namely  averaging and sum.

The authors have explored well the different families of methods designed for  learning on graphs; ranging from spectral to spatial methods. In this paper, the proposed method can be added to the class of spatial methods. Relevant works to the contribution are discussed and cited in the related works parts and experiments.

On of the strong points of the paper is the anchor representation which enables the explicit design of anisotropic convolution operator. This representation is based on K-means sampling. It is beneficial to control the variable-sized graphs.

However, the proposed network is relatively expensive in time and space which can hinder processing very large graphs and datasets. For instance, in community detection and particle physics simulations where the number of nodes is in general higher than 2 millions. The latter can be reduced drastically to thousands of nodes thanks to subsampling techniques such as K-means so that Graph Deformer Network can be tractable in practice at the expense of a possible loss of geometric structure. The idea is to come up with a trade-off between the number of nodes with respect to the initial geometry and the expressive power of the anisotropic convolutional operator.

In order to clarify some points, l have some questions :

1- How the problem of over-smootheness is handled  by the GDN?

2- In equation (10), why only second order is considered ? what about the diameter of the graphs ? or different orders w.r.t to the graph connectivity and edge distribution at a given neighboring systems.

3- How different is the injective function proposed in this work compared to the one in GIC network ? It is mentioned in the supplementary part  that the designed injective function is stronger than standard schemes based on averaging and sum aggregations. However, in Table 2 (second column), GIC outperforms GDN.

4- Why in the setting of graph classification, GDN is composed of 3 layers while in node classification setting, GDN is composed of 2 layers ? even  if node classification data are larger than graph classification.

5- How do you explain the slight gain in performances for GDN with pubmed dataset compared to other dataset ?

6- Pubmed data is composed of very large graphs, how do you explain the fact that it needs only 2 layers as other datasets ? It could be interesting to show the results with different number of layers to see whether the depth of the network brings extra gain or suffers from over-fitting. Moreover, it’s still not clear whether deeper networks are needed to tackle graph data. For that reason, it could be interesting to summarize the depth and width of the proposed models for each dataset for more interpretability. The depth can be upper-bounded by the diameter of the graph in the case of symmetric graphs.
7- How could one explain that the sum of neighbors in the local neighborhood is the same as the sum on each anchor node after deformation?

For more clarity tables could be organized as follow :

A- Baselines (Linear SVM, MLP)

B- Euclidean ConvNets

C- Non Euclidean ConvNets (C-1 : Spectral methods , 2- : Spatial methods)

As future works, one may consider :

A-  To study the proposed network from spectral standpoint and visualize the Fourrier atoms  to analyze the behaviour of  the anisotropic convolutional operation.

B- To show the genericity of the proposed network, it could be beneficial to extend this work to challenging real world problems such as spatio-temporal data in computer vision, high energy physics,... etc. Moreover, node regression could be considered.

C- Linear SVM and MLP classifiers could be a very important baselines. The latter allow to assess  the extra gain of GCN with respect to the number of parameters of the model.

The approach is well placed in the litterature, the organization of the paper is clear and the supplementary part is helpful including theoretical details and experimental ones. However, the experimental part could be explored more.

For all these reasons, l would like to recommend to accept this paper. Beneficial for both theoretical follow-up studies and empirical ones.

---

### Official Review · AnonReviewer4 · 2020-11-03
**Review for Graph Deformer Networks**

**Rating:** 5
**Confidence:** 5

**Review:**

This work tries to realize the anisotropic convolution by using anchor nodes in the graph. There are some concerns for this work:

1.	The key idea of space conversion is to use the attention operator to generate a fixed number of features for each node. The anchor nodes features are serving as query vectors and produce the output by attending each node’s neighboring nodes. In this way, each node will have a fixed number of feature representations. However, the regular convolution cannot be applied due to lack of locality information. In the paragraph under Eq. (8), the authors claimed that this can be viewed as a regular 2-D plane. Since there is still no ordering information among these selected anchors, how can we view them as a regular 2-D plane. Similar argument presents in section 2.3.2. Currently, I cannot see a solid support for this argument, which directly leads the failure of motivation in this work.

2.	Permutation invariant issue exists in this work. We know that the permutation invariant is an important property especially for GNN methods. However, in Eq. (3), the clustering algorithm is used to determine the anchor vectors based on sampled nodes. This means different execution of the algorithm will lead to different results. The authors need to clarify this.

3.	The authors perform extensive experimental studies on various datasets. However, I would recommend the authors to provide results on relatively larger datasets such as REDDIT-MULTI-12K. Also, some numbers reported in Table 2 are not consistent with those in the baseline papers such as GIN. The baseline models are quite outdated. There have been many works in this field such as StrctPool.

---

### Decision · Program_Chairs · 2021-01-07
**Final Decision**

**Decision:**

Reject

**Comment:**

Three of the reviewers are significantly concerned about this submission, while R1 is positive. Meanwhile, R1 is also concerned about some details in the paper, including space and time complexity etc. The authors provided detailed feedback to these comments, but R1 does not provide support to this work during discussions. Thus a reject is recommended.